# Prevalence and Predictors of Antibiotic Self-Medication in Sudan: A Descriptive Cross-Sectional Study

**DOI:** 10.3390/antibiotics12030612

**Published:** 2023-03-20

**Authors:** Mohamed A. Hussain, Ahmed O. Mohamed, Omalhassan A. Abdelkarim, Bashir A. Yousef, Asma A. Babikir, Maysoon M. Mirghani, Entsar A. Mohamed, Wadah Osman, Ramzi A. Mothana, Rashid Elhag

**Affiliations:** 1Department of Pharmaceutical Microbiology, Faculty of Pharmacy, International University of Africa, Khartoum P.O. Box 2469, Sudan; 2Department of Pharmacy Practice and Clinical Pharmacy, Faculty of Pharmacy, International University of Africa, Khartoum P.O. Box 2469, Sudan; 3Department of Pharmacology, Faculty of Pharmacy, University of Khartoum, Khartoum P.O. Box 1996, Sudan; 4Department of Pharmaceutical Microbiology, Pharmacy Program, Al-Yarmouk College, Khartoum P.O. Box 11111, Sudan; 5Department of Pharmacognosy, Faculty of Pharmacy, University of Khartoum, Khartoum P.O. Box 1996, Sudan; 6Department of Pharmacognosy, College of Pharmacy, King Saud University, Riyadh 11451, Saudi Arabia; 7Department of Biology, College of Science and Technology, Florida A & M University, Tallahassee, FL 32307, USA

**Keywords:** antibiotic self-medication, Sudan, antimicrobial resistance

## Abstract

Background: Self-medication with antibiotics (SMA) is one of the common factors which precipitate antimicrobial resistance, yet if effective implementations are amended it can be effortlessly controlled. The present study aimed to estimate the prevalence and predictors of SMA in Sudan. Methods: The study adopted a cross-sectional study design conducted in all Sudan states between June and December 2021. Multi-stage stratified cluster sampling was used. A semi-structured questionnaire was used for data collection. Descriptive statistics were used to present the data. Binary logistic regression was computed to investigate the possible factors which associated with SMA. Results: Out of 1492 participants surveyed, 71.3% utilize antibiotics as self-medication. The derived reasons for SMA were convenience (63.3%) and cost-saving (34.8%). Tonsillitis was the most common ailment behind SMA (55.5%). Log-binominal regression revealed that non-insured and low level of education participants were more likely to predict SMA. Regarding the practice, 40% changed the dose and/or antibiotics mainly owing to improvement (53.7%) or worsening of the condition (37.9%). The most commonly used antibiotic was amoxicillin/clavulanic acid (32.5%). Conclusions: Two out of three individuals in Sudan practice SMA mainly to manage upper respiratory tract ailments. Thus, the necessity of implementing an antimicrobial stewardship program throughout the country, as well as implementing effective legislation to prohibit dispensing antibiotics without prescription is urgently required.

## 1. Introduction

The behaviors of patients toward different medical conditions vary considerably, from immediately seeking medical advice, relying on self-medication (SM), or neglecting the condition [1]. The World Health Organization (WHO) described self-medication (SM) as the use of medicinal products or herbs to manage self-diagnosed disorders or symptoms; moreover, SM also comprises repeated or continued use of a prescribed drug for chronic, recurrent diseases or symptoms. This generally occurs through obtaining medicines without a prescription, sharing medicines, or using leftover medicines stored at home [2].

Even though SM might have some positive outcomes when used properly such as reducing the cost of the treatment [3], numerous pitfalls are associated with the inappropriate use of SM including: delay in the treatment, drug–drug interactions, masking of symptoms, adverse drug reactions (ADR), and most importantly antimicrobial resistance (AMR) [3,4]. AMR was affirmed by the WHO as one of the major problems facing humanity [5]. Overuse and misuse of antibiotics through inadequate dosing, incomplete dose, extensive veterinary use, public beliefs that antibiotics can cure any conditions, overprescribing, non-prescription, and self-medication with antibiotics (SMA) are common factors that are associated with the development of AMR [6,7]. SMA in low to middle-income countries (LMICs) is thought to be more prevalent, mainly owing to the combined effect of external factors such as dispensing antibiotics without prescription and internal factors like economic status [6,8].

Worldwide, effective implementation of rational antibiotic prescription is lacking despite the availability of issued legislation. This is quite understandable when you come to know that two-thirds of antibiotics are available without prescription in the pharmaceutical sector as stated by WHO [9]. Additionally, a recent study estimated that half of the antibiotics were purchased without prescription globally, the same study outlined that non-prescription use of antibiotics reached 82% in some middle-eastern countries [10]. In LMICs, the SM and inappropriate antibiotic practice was declared to be more intensive, according to a meta-analysis published in 2021, in which SMA was outlined to be ranging between 50–93.8% [8]. Another meta-analysis review reported 55.7% as an overall median prevalence of SM with antibiotics in Africa [11]. The same study pointed to the highest prevalence of SMA identified in the west and north African sub-regions [11].

Antibiotics in Sudan are listed under prescription only medicine, albeit the presence of clear regulations, which prohibit dispensing of antibiotics as over the counter drug, antibiotics in Sudan like its counterpart from developing countries can be accessed easily. Most of the antibiotics were purchased from pharmacies in several regions in Sudan [12,13,14]. This is compatible with the preceding report which revealed that more than 80% of the pharmacists in Sudan frequently dispense antibiotics without prescription [15].

A study conducted in Khartoum state showed that more than 80% of the communities were self-medicated, among them approximately one-third (28.7%) were antibiotic utilizers [13]. The prevalence of SMA in Khartoum state has been described by Abdelmoneim Awad et al. to be 73.9% in 2005 [16]. Another recent study reported a prevalence of 60.3% among Sudanese undergraduate medical students [17]. Most of the studies which were carried out in Sudan pointed toward the association of SMA with age, gender, income, and level of education [13,16,17,18]. It is worth mentioning that penicillin antibiotics were declared to be the most commonly prescribed antibiotics in most of the studies [17,19].

Given the above backdrop, the present study aimed at providing an updated and comprehensive nationwide estimation of the prevalence of SMA in general Sudanese communities and its association with socio-demographic factors, also the study investigated predictors of SMA including the most common reasons and ailments behind SMA and the most frequently self-medicated antibiotics.

## 2. Results

### 2.1. Demographic Characteristics of the Included Participants

Out of 1492 participants surveyed in this study from all Sudan states (The distribution of the participants according to their states is provided in Appendix A), 53.4% (796) were female. The majority of the participants were aged from 18 to 24 (36.3%) and 25 to 39 (38.7%). More than half (54.3%) of the participants had a low monthly income, however, more than two-thirds (69.3) were medically insured. Regarding the level of education, more than half (55.9%) of the participants completed their university studentship. The demographic characteristics of the included participants are presented in Table 1.

### 2.2. Prevalence, Sources, and Reasons behind SMA

More than two-thirds (71.3%) of the participants used antibiotics as SM. The Chi-square test revealed that, the participants level of education was significantly associated with antibiotic SM. The vast majority of the participants obtained antibiotics from the pharmacy (92.1%). Additionally, graduate (93.9%) and post-graduate (94.4%) participants obtained antibiotics from pharmacies in higher proportions than primary (89.2%) and secondary school (86.5%) participants, Table 2.

### 2.3. Common Ailments for Taking Antibiotic as SM

Tonsillitis was the most common aliment that drove participants to self-treatment. It was rated by more than half of the participants (55.5%). This was followed by cough (45%), while a small proportion of the participants used antibiotics for vomiting (10%) (Table 3).

### 2.4. Factors Associated with the Use of Antibiotics as SM

At the bivariate level, medically insured (COR: 0.656; 95% CI (0.506–0.817) were more likely to use antibiotics as SM. Individuals having secondary (COR: 0.61; 95% CI (0.39–0.97), and post-graduate (COR: 0.38; 95% CI (0.23–0.63) levels of education were less likely to use antibiotics as SM, other factors were not statistically significant. Regarding the multivariate model, the model sensitivity was 70.2%, further, the model adequately fits the data since there were no differences between the observed and the predicted (Hosmer and Lemeshow test = 0.761). Medically non-insured (AOR: 0.645; 95% CI (0.487–0.855) and post-graduates (AOR: 0.27; 95% CI (0.15–0.5) were the only predictors for SMA. Complete logistic regression for the use of antibiotic as SM are presented in Table 4.

### 2.5. Knowledge and Adherence to Antibiotic Dosage

More than half of the participants knew the dosage of antibiotics through pharmacist consultation (56.4%) and they fully understood the instructions (59.6%). About 40% of the participants sometimes changed the dosage and/or the antibiotics deliberately during treatment, while 10% always changed the dosage. The main reasons for changing the dosage were improvement (53.7%) or worsening of the condition (37.9%). Approximately two-thirds (67.1%) of the participants changed the former antibiotics if they weren’t effective, on the other hand, more than half of the participants (55.1%) stopped taking the antibiotics when symptoms disappeared, and about 10% consulted the doctor or the pharmacist before stopping the antibiotics (Table 5).

### 2.6. Commonly Used Antibiotics and Common Adverse Reactions

The most commonly used antibiotics were amoxicillin/clavulanic acid combinations (32.5%), followed by amoxicillin (26.5%), metronidazole (25.3%), and azithromycin (25.3%). About one-fifth (21.8%) of the participants experienced ADR when they took antibiotics. The primary action taken by more than half (57%) of those who experienced ADR was to stop the antibiotics (Table 6). About half of the participants thought that SMA (51.9%) was not an acceptable practice, and more than one-third (37.4%) thought that they cannot treat the infectious disease on their own. Most of the participants (61%) selected antibiotics based on its indications, and about half of the participants used antibiotics when recommended by community pharmacists (47.9%) and according to their own experience (46.8%). The type of antibiotics was the main factor considered by the majority of the participants (45.4%), while the brand of the antibiotics was only considered by less than one-fifth of the participants (16.3%) (Table 6).

## 3. Discussion

SMA is one of the common factors which precipitate AMR, yet if effective implementations are adopted it can be easily controlled. Tracking SM behaviors of public individuals are of paramount importance since it facilitates the development of preventable measures towards this condition. Moreover, it also uncovers the weakest domains in the health system. It was on these grounds that the current nationwide survey was conducted to determine the prevalence of SMA in different Sudanese states, as well as to provide an insight into the reasons, and factors associated with SMA.

The current cross-sectional survey indicated that the prevalence of SMA among the Sudanese community was 71.3%. The reported figure lies in the middle of local and regional figures in previous studies. For instance; locally, one of the earliest and most comprehensive surveys conducted by Abdelmoneim Awad et al. in 2005 reported a prevalence of 73.1% [16], while a recent study which included Sudanese university medical students reported that antibiotics were self-medicated by 60% of the students [17]. Regionally, a meta-analysis review pooled the prevalence of SMA in Africa using 40 studies from 19 countries and the computed prevalence ranged from 50–93.8% [11]. On the other hand, another systematic review outlined the proportion of SMA in middle-eastern countries to be in the range of 12.1–93.1% [10]. Our finding points towards a higher prevalence of SMA in Sudan, unfortunately, it also emphasizes the fact that the practice of SMA in Sudan has remained consistent throughout the last 20 years.

Different reasons behind SMA were mentioned previously, such as cost saving, previous experience, and convenience which were reported repeatedly. However, less frequently, the emergence of illness, and the long delays in clinic have been reported [1,20,21,22]. Likewise, participants in the present study cited convenience (63.3%) and cost-saving (41%) as common reasons for SMA (Table 2). The finding is consistent with the results reported previously in Sudan, Malaysia, India, and Ethiopia [14,23,24,25].

The present study illustrated the common ailments for SMA. Generally, SMA to manage upper respiratory tract infections (URT) such as tonsillitis (55.5%), cough (45%), runny nose (37.8%), and nasal decongestion (33.4%) were higher than other ailments which comprise fever (31.9%), pain (29.1%), diarrhea (21.9%), and wound infection (14.9%) (Table 3). This finding is in line with previous studies conducted in Sudan, Tanzania, and India [12,21,23]. Bearing in mind that most of the URT infections are of viral origin, and antibacterial agents must be preserved only for bacterial infections which indeed requires a series of investigations and diagnoses provided by health care specialists, and considering that this pattern remains consistent in Sudan through the last 20 years with a gradual increment, health authorities in Sudan should effectively implement an antimicrobial stewardship program to optimize the utilization of antimicrobial agents.

Health services in Sudan are provided by different bodies including: government, private sectors, army, police, universities, and civil society [26]. The national health insurance fund (NHIF) is an extension of social health insurance which was introduced in 1994, the finance of NHIF is based on cost sharing (national social system based on the cooperation between the government and community) [27]. The coverage in all states is around 50% (except Khartoum = 70%), and the out of pocket share in Sudan is reported to be 70% [27,28]. Nearly one third (30.7%) of the participants from the present study were not medically insured (Table 1). Further, insurance status was significantly associated with SMA (*p*-value < 0.00), binary logistic regression indicated that medically non-insured participants were less likely to use antibiotics as SM in comparison to insured participants (COR: 0.656; 95% CI (0.506–0.718), (AOR: 0.645; 95% CI (0.487–0.855) (Table 4). Similar findings have been reported previously in Pakistan [12]. Additionally, 41% declared cost saving as one of the main reasons behind SMA. On the other hand, participants with secondary school and post-graduates were less likely to take SMA compared to primary school levels of education (Table 4). This pattern is not limited to this study, and it has been observed in previous studies in Lebanon, Uganda, and Malaysia [18,29,30]. However, it contradicted studies carried out in Sudan, Eritrea, and Bangladesh [16,20,31]. Such a finding is best explained by the fact that educated individuals understand the difficulties in discriminating infectious diseases and knew the consequences of SMA, therefore, they prefer to visit doctors instead of self-medicating.

Previous studies outlined that the main focus of community pharmacists in Sudan is to efficiently prescribe medications [32,33]. A considerable proportion of the participants from the present study sought antibiotics mainly from the community pharmacies (90%), and the remaining participants obtain antibiotics from leftover medication (Table 5). Previous researchers in Sudan reported a similar pattern [14,17]. This finding indicated that the gap between the actual role of the community pharmacist which is extended to include patient counseling and education is a promising area for mitigating SMA.

Additionally, more than half of the participants in this study (51.9%) thought that SMA is not an acceptable practice (Table 5 and Table 6). Paradoxically, the practice of the participants diverges from rationality, when you come to know that 41% of the participants change the dosage of the antibiotics deliberately (Table 5). Moreover, the fact that a high percentage of the enrolled participants switch antibiotics harmonizes with the finding that only 37% stop taking antibiotics after dosage completions. The abovementioned malpractice is consistent with previous studies conducted in India (24%), Malaysia (41%), and Egypt (71%) that participants change the dosage of antibiotics during usage [23,24,34]. In Afghanistan, 33% of the participants stop taking antibiotics [35], while in Malaysia 35.3% of university students switch the dosage of antibiotics [24]. Given the above backdrop, it is not surprising that at the national level multi drug resistant and extensively drug-resistant isolates detected from clinical specimens are increasingly reported [7,36,37].

Participants in the current study cited amoxicillin/clavulanic acid as the most common antibiotic used as SM (32.5%), followed by amoxicillin (26.5%), azithromycin, and metronidazole (25.3%) (Table 6). Similar results were observed previously in Sudan [14,16], where azithromycin (29.9%) and amoxicillin/clavulanic (26.8%) were found to be the most common antibiotics self-medicated by university students [17]. Multiple studies in Africa and the Middle-East concluded the extensive use of beta-lactam antibiotics especially amoxicillin and amoxicillin/clavulanic as SM [10,11]. It is however, worth mentioning that earlier studies in Sudan reported amoxicillin as the most common antibiotic used in comparison to the present study and a recent study in 2022 [12,16]. This shift can be explained by the fact that patients always seek the most effective antibiotics, or it might be due to extensive promotion applied by different companies to promote their antibiotics (amoxicillin/clavulanic).

Besides accelerating antimicrobial resistance, SMA can also be associated with ADR. One-fifth (21.8%) of study participants reported that they experienced ADR (Table 6). This is slightly lower than a previous study in Malaysia (28.3%) [5]. Alarmingly, a considerable amount of the participants either switched the antibiotics or continued the antibiotic with the rate of 20.8% and 11.3%, respectively. ADRs associated with antibiotics ranged from mild side effects such as GIT symptoms to life-threatening conditions such as anaphylactic shock which is associated with a large number of antibiotics impacting patients’ health as well as cost [25,38].

The finding from the present study can be partially generalized to the overall Sudanese community owing to the large and diverse sample size. However, one of the limitations of this study was the recall bias since not all participants were able to exactly remember for instance the types of antibiotics. Further, the study was subjected to selection bias, since it was conducted during the daytime in public areas, it is for this reason most of the participants were aged below 39 years old. Additionally, the questionnaire used in the present study adopted close-ended limited options which made it difficult for some respondents to express their opinions.

## 4. Materials and Methods

### 4.1. Study Design and Setting

The study adopted a cross-sectional descriptive study design, conducted in all Sudan states (all 18 states) through the period between 1 June and 15 December 2021.

### 4.2. Study Population

All Sudanese adult aged above 18 years old and willing to participate in the study were considered eligible.

### 4.3. Sample Size and Sampling Technique

According to the last census, the total population of Sudan is around 46,000,000. Using the formula below:n = Z^2^ p (1 − p)/w^2^
where n: sample size, Z: the critical vale (using confidence interval of 99% (Z = 2.326)), p: proportion of the target population estimated to have a particular characteristic (since there were no previous nationwide study the frequency of occurrence was assumed to be, p = 50%), (1 − p): (frequency of not occurrence of an event), w: desired margin of error tolerated (degree of precision, w = 4%). Thus, the calculated sample size was 1041 participants. We collected data from 1492 accounting for missing data. A multistage stratified sampling technique was applied to the participants. Sudan was divided into 18 states. Each state was considered a stratum, and then within each stratum, participants were selected randomly using a convenience sampling technique. Samples were collected from public places such as markets, parks, and bus stations.

### 4.4. Operational Definitions and Study Variables

SMA (dependent variables) was defined as the selection and use of antibiotics by participants, within the last 12 months, to manage at least one self-recognized illness or symptom without professional prescription and supervision regarding indication, dosage, and duration of treatment. Independent variables (predictors of SMA) were carefully selected based on previous studies, including: participants’ gender, age, monthly income, insurance status, and educational status (all were categorical variables).

### 4.5. Data Collection

The current study used semi-structured questionnaires for data collection (participants who found difficulties in writing were interviewed by the trained data collectors based on the questionnaire). A comprehensive search of the literature for potential studies reporting SMA was carried out through different databases to get guidance in designing the questionnaire [20,23,24,36]. The questionnaire consisted of 25 items (provided in Appendix A), which can be broadly divided into two main sections; the first section gives information regarding the demographical characteristics of the study participants, which includes gender, age, economic status, insurance status, and levels of education, while the second section starts with a main question which seeks information about any previous use of antibiotics without prescription in the last 12 months through a closed-end format (yes/no). Participants whose answer is ‘yes’ in the previous question were further asked to explain the main reasons and major ailment that led participants to self-medicate (multiple choice questions). It also emphasizes the practice of the participants through enquiring about the sources, selection, and adherence to antibiotic regimens (closed and close multiple choice questions). Furthermore, commonly used antibiotics and adverse drug reaction histories were also reported (closed and close multiple choice questions). For the purpose of validation, two experts in pharmacy practice were asked to highlight the main weakness of the developed questionnaire, and their comments were considered in the final version. Additionally, a pilot study was distributed to 20 individuals to confirm the clarity of the questions, the questionnaire was further validated through Cronbach alpha (α = 0.78). Responses from the pilot study were excluded from the study. To ensure the quality of the data; data was collected only through trained fifth-year pharmacy students who were taught courses in research methodology and given a comprehensive presentation on the research topics. Furthermore, data collectors were asked to check the completeness of each questionnaire.

### 4.6. Data Analysis

Data were entered into a Microsoft Excel spreadsheet, coded, and exported to the statistical software package SPSS (version 25.0). Both descriptive and inferential statistics were used to analyze the data. The main parts of the questionnaire were expressed in terms of frequency and percentage. A Chi-square test was employed to study the relationship between socio-demographic factors and other variables. A binary logistic regression model was used to assess the association between prevalence SM and explanatory variables. Regardless of their *p*-value in the unadjusted analysis, all variables were included in the final multiple regression model, and the model appropriateness was tested using Hosmer and Lemeshow test. Both crude odds ratio (COR) and adjusted odds ratios (AOR) were reported with a 95% confidence interval (95% CI). Finally, a *p*-value less than 0.05 was considered significant.

### 4.7. Ethical Consideration

Ethical approval was obtained from the ethical committee at the University of Khartoum, Faculty of Pharmacy (FPEC-26-2021). Before conducting the study, all participants signed written informed consent after a clear explanation of the research objectives, and each participant had the right to withdraw at any time from the study. To ensure confidentiality, all questionnaires were coded and personal identifiers remained anonymous throughout the study.

## 5. Conclusions

Two out of three individuals in Sudan SMA mainly to manage URT ailments, this mal-practice was explained by most of the participants by it is convenience and cost-saving. Amoxicillin/clavulanic were the most commonly used antibiotics. SMA was associated with participant’s level of education and insurance status. The findings from the present study indicate the necessity of activating antimicrobial stewardship programs throughout the country, as well as implementing effective legislation to prohibit dispensing antibiotics without prescription.

## Figures and Tables

**Table 1 antibiotics-12-00612-t001:** Demographic characteristics of the included participants and their association with the use of antibiotic as self-medications.

Demographic Characteristics	Frequency (%)	I Use Antibiotic as Self-Medication (%)	χ^2^ *p*-Value
Gender			
Male	695 (46.6)	523 (72)	0.71
Female	796 (53.4)	590 (71)	
Age (years)			
18–24	542 (36.3)	374 (69)	0.438
25–39	577 (38.7)	422 (73)	
40–59	303 (20.3)	221 (73)	
More than 60	70 (4.7)	45 (64)	
Monthly income ($)			
Less than 50	810 (54.3)	575 (71)	0.089
50–99	297 (19.9)	220 (74)	
100–149	149 (10)	104 (70)	
More than 150	236 (15.8)	182 (77)	
Insurance status			
Insured	1034 (69.3)	745 (72)	>0.00
Non-insured	458 (30.7)	285 (62.3)	
Level of education			
Primary school	146 (9.8)	114 (78)	>0.00
Secondary school	352 (23.6)	243 (69)	
Graduate	834 (55.9)	617 (74)	
Post-graduate	160 (10.7)	91 (57)	
Total	1492 (100)	1059 (71.3)	

Legends: (1 United State Dollars = 250 Sudanese Pounds, all conversion were made based on the central bank of Sudan).

**Table 2 antibiotics-12-00612-t002:** Relation between demographical characteristic of the included participants and common sources and reasons behind SMA.

	Source of Antibiotics		Reason behind SMA		
Demographic Characteristics	Pharmacy (%)	Left-Over (%)	Cost Saving (%)	Convenience (%)	Lack of Trust in Prescribing Doctor (%)
Gender					
Male	472 (94.8)	97 (19.5)	230 (46.2)	294 (59.0)	110 (22.1)
Female	505 (90.2)	150 (26.8)	204 (36.4)	379 (67.7)	93 (16.6)
Age (years)					
18–24	338 (90.1)	89 (23.7)	131 (34.9)	252 (67.2)	70 (18.7)
25–39	389 (93.1)	100 (23.9)	191 (45.7)	257 (61.5)	84 (20.1)
40–59	209 (94.6)	46 (20.8)	89 (40.3)	141 (63.8)	36 (16.3)
More than 60	41 (93.2)	12 (27.3)	23 (52.3)	23 (52.3)	13 (29.5)
Monthly income ($)					
Less than 50	516 (91.8)	129 (23.0)	245 (43.6)	356 (63.3)	92 (16.4)
50–99	202 (93.5)	63 (29.2)	88 (40.7)	131 (60.6)	48 (22.2)
100–149	95 (92.2)	23 (22.3)	33 (32.0)	71 (68.9)	24 (23.3)
More than 150	164 (92.1)	32 (18.0)	68 (38.2)	115 (64.6)	39 (21.9)
Insurance status					
Insured	703 (92.3)	184 (24.1)	307 (40.3)	494 (64.8)	201 (26.4)
Non-insured	274 (92.3)	63 (21.2)	127 (42.8)	179 (60.3)	73 (24.6)
Level of education					
Primary school	102 (90.3)	33 (29.2)	64 (56.6)	61 (54.0)	22 (19.5)
Secondary school	205 (85.1)	64 (26.6)	116 (48.1)	147 (61.0)	41 (17.0)
Graduate	582 (94.8)	125 (20.4)	217 (35.3)	412 (67.1)	116 (18.9)
Post-graduate	88 (97.8)	25 (27.8)	37 (41.1)	53 (58.9)	24 (26.7)
Total	274 (92.3)	247 (23.3)	434 (41.0)	673 (63.6)	22 (19.5)

**Table 3 antibiotics-12-00612-t003:** Relation between common ailments of SMA and demographic characteristic of the included participants.

	Tonsillitis	Cough	Runny Nose	Nasal Congestion	Fever	Pain	Diarrhea	Wound Infection	Vomiting
Gender									
Male	274 (55)	222 (44.6)	207 (41.6)	166 (33.3)	147 (29.5)	127 (25.5)	109 (21.9)	85 (17.1)	48 (9.6)
Female	314 (56.1)	255 (45.5)	193 (34.5)	188 (33.6)	191 (34.1)	181 (32.3)	123 (22)	73 (13)	60 (10.7)
Age (years)									
18–24	180 (48)	161 (42.9)	126 (33.6)	126 (33.6)	127 (33.9)	110 (29.3)	67 (17.9)	41 (10.9)	35 (9.3)
25–39	263 (62.9)	189 (45.2)	157 (37.6)	133 (31.8)	122 (29.2)	109 (26.1)	85 (20.3)	73 (17.5)	37 (8.9)
40–59	123 (55.7)	104 (47.1)	94 (42.5)	77 (34.8)	67 (30.3)	67 (30.3)	58 (26.2)	35 (15.8)	24 (10.9)
More than 60	22 (50)	23 (52.3)	23 (52.3)	18 (40.9)	22 (50)	22 (50)	22 (50)	9 (20.5)	12 (27.3)
Monthly income ($)									
Less than 50	291 (51.8)	245 (43.6)	195 (34.7)	179 (31.9)	199 (35.4)	182 (32.4)	116 (20.6)	71 (12.6)	55 (9.8)
50–99	132 (61.1)	93 (43.1)	81 (37.5)	81 (37.5)	62 (28.7)	64 (29.6)	68 (31.5)	43 (19.9)	28 (13)
100–149	60 (55.3)	59 (57.3)	49 (47.6)	34 (33)	34 (33)	29 (28.2)	22 (21.4)	18 (17.5)	9 (8.7)
More than 150	105 (59)	80 (44.9)	75 (42.1)	60 (33.7)	43 (24.2)	33 (18.5)	26 (14.6)	26 (14.6)	16 (9)
Insurance status									
Insured	426 (55.9)	351 (46.1)	291 (38.2)	274 (36.0)	242 (31.7)	220 (28.9)	166 (21.8)	109 (14.3)	426 (55.9)
Non-insured	162 (54.7)	125 (42.2)	109 (36.7)	80 (26.9)	97 (32.5)	87 (29.4)	66 (22.1)	49 (16.6)	162 (54.7)
Level of education									
Primary school	46 (40.7)	53 (46.9)	54 (47.8)	40 (35.4)	49 (43.4)	23 (20.4)	40 (35.4)	16 (14.2)	16 (14.2)
Secondary school	132 (54.8)	102 (42.3)	95 (39.4)	88 (36.5)	80 (33.2)	66 (27.4)	50 (20.7)	34 (14.1)	35 (14.5)
Graduate	350 (57)	282 (45.9)	219 (35.7)	194 (31.6)	201 (32.7)	189 (30.8)	134 (21.8)	88 (14.3)	53 (8.6)
Post-graduate	60 (66.7)	40 (44.4)	32 (35.6)	32 (35.6)	8 (8.9)	30 (33.3)	9 (10)	20 (22.2)	4 (4.4)
Total	588 (55.5)	477 (45)	400 (37.8)	354 (33.4)	338 (31.9)	308 (29.1)	232 (21.9)	158 (14.9)	108 (10.2)

**Table 4 antibiotics-12-00612-t004:** Bivariate and multiple logistic regression of the demographic factors of SMA.

Demographic Characteristics	Crude Odds Ratio (95% CI)	*p*-Value	Adjusted Odds Ratio (95% CI)	*p*-Value
Gender				
Male	1 (baseline)		1 (baseline)	
Female	0.956 (0.761–1.20)	0.700	0.997 (0.758–1.321)	0.984
Age (years)				
18–24	1 (base line)		1 (baseline)	
25–39	1.178 (0.908–1.529)	0.217	1.067 (0.767–1.485)	0.669
40–59	1.184 (0.865–1.621)	0.293	1.04 (0.689–1.571)	0.852
More than 60	0.784 (0.465–1.323)	0.362	0.687 (0.353–1.337)	0.269
Monthly income ($)				
Less than 50	1 (baseline)		1 (baseline)	
50–99	1.195 (0.868–1.643)	0.274	1.3 (0.892–1.894)	0.172
100–149	0.959 (0.641–1.436)	0.840	0.952 (0/59–1.537)	0.841
More than 150	1.333 (0.932–1.907)	0.166	1.976 (1.244–3.14)	0.004 *
Insurance status				
Insured	1 (baseline)		1 (baseline)	
Non-insured	0.656 (0.506-0.817)	0.000 *	0.645 (0.487-0.855)	0.002 *
Level of education				
Primary school	1 (baseline)		1 (baseline)	
Secondary school	0.618 (0.393–0.972)	0.037 *	0.658 (0.39–1.111)	0.117
Graduate	0.786 (0.516–1.199)	0.264	0.721 (0.44–1.181)	0.193
Post-graduate	0.382 (0.23–0.633)	0.000 *	0.278 (0.152-0.508)	0.000 *

Legends: CI = confidence interval, * = significant *p*-Value.

**Table 5 antibiotics-12-00612-t005:** Knowledge, practice, and adherence to dosage of antibiotics and/or instructions.

Source and/or Adherence Practice	Practice	Frequency (%)
Did you ever check the instructions that come with the package insert of antibiotics for self-treatment	Always	514 (48.5)
	Sometimes	323 (30.5)
	Never	222 (21)
How much did you understand the instructions that come with the package insert of antibiotics for self-treatment	Fully understood	631 (59.6)
	Partly understood	381 (36)
	Did not understand at all	47 (4.4)
How did you know the dosage of antibiotics ^a^	By checking the package insert	371 (35)
	By consulting a doctor	304 (28.7)
	By consulting a pharmacist	597 (56.4)
	By consulting family members/friends	131 (12.4)
	From the Internet	93 (8.8)
	From my previous experience	210 (19.8)
	By guessing the dosage by myself	54 (5.1)
Did you ever change the dosage of antibiotics deliberately during the course of self-treatment	Always	114 (10)
	Sometimes	445 (42)
	Never	500 (47.2)
Why did you change the dosage of antibiotics during the course of self-treatment ^a^	Improving conditions	300 (53.7)
	Worsening conditions	212 (37.9)
	To reduce adverse reactions	101 (18.1)
	Drug insufficient for complete treatment	100 (17.8)
Did you ever switch antibiotics during the course of self-treatment	Always	87 (8.2)
	Sometimes	430 (40.6)
	Never	542 (51.2)
Why did you switch antibiotics during the course of self-treatment ^a^	The former antibiotics weren’t effective	393 (67.1)
	The latter one was cheaper	85 (16.5)
	To reduce adverse reactions	121 (23.4)
	Based on my experience	64 (12.3)
Have you ever found out that you had taken the same antibiotics with different names at the same time	Yes	632 (59.7)
	No	427 (40.3)
When did you normally stop taking antibiotics	After a few days regardless of the outcome	212 (20)
	After symptoms disappeared	501 (55.1)
	A few days after the recovery	291 (29.1)
	At the completion of the course	401 (37.9)

Legends: ^a^ = more than one options is allowed.

**Table 6 antibiotics-12-00612-t006:** Commonly used antibiotics and common adverse reactions.

Commonly Used Antibiotics and Adverse Reactions	Practice	Frequency (%)
Commonly used antibiotics	Amoxicillin and clavulanic acid	344 (32.5)
	Amoxicillin	281 (26.5)
	Metronidazole	268 (25.3)
	Azithromycin	268 (25.3)
	Don’t remember	236 (22.3%)
Your selection of antibiotics was based on	Indications	646 (61)
	Recommendation by community pharmacists	507 (47.9)
	My own experience	496 (46.8)
	Opinion of family members	231 (21.8)
	Opinion of friends	230 (21.7)
What did you consider when selecting antibiotics	Previous doctor’s prescription	205 (19.4)
	Type of antibiotics	481 (45.4)
	Price of antibiotics	229 (21.6)
	Brand of antibiotics	173 (16.3)
Have you ever had adverse drug reaction when taking antibiotics	Yes	231 (21.8)
	No	828 (78.2)
What did you do for adverse drug reaction	Stop taking the antibiotics	131 (57.7)
	Switch the antibiotics	48 (20.8)
	Consulted a pharmacist	64 (27.6)
	Consulted a doctor	55 (24)
	Consulted family members/friends	26 (11.3)
	No action	35 (13.2)
What do you think about self-medication with antibiotics for self-health care	Good practice	132 (12.5)
	Acceptable practice	377 (35.6)
	Not acceptable practice	550 (51.9)
Do you think you can treat common infectious diseases with antibiotics successfully by yourself	Yes	164 (15.5)
	No	396 (37.4)
	Not sure	499 (47)

## Data Availability

Most of the relevant data are available in the main text, further data are available from the corresponding author upon reasonable request.

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
