# Peer review of "Prevalence and Predictors of Antibiotic Self-Medication in Sudan: A Descriptive Cross-Sectional Study"

_antibiotics, 2023, doi:10.3390/antibiotics12030612_

Round 1

Reviewer 1 Report

The manuscript report results of a nationwide study conducted in Sudan, Africa, about prevalence of antibiotic self-medication (ASM) and the factors associated.

The study can potentially contribute to understanding this behavior, which has consequences of paramount importance for antimicrobial resistance and public health.

Despite this, there are complaints related to Methodology and Results presentation that need revision.

Major

Materials and Methods

Sampling technique:  Please explain how the random sampling technique was applied (e.g., source of all participants from which the sample was drawn)

Data collection:

- Give details about which section of the questionnaire was self-administrated. Did the questions on SMA self-administered? What questions?

- How many interviewers conducted the survey? Were they trained to collect data (including written protocol or operation manual)? Explain if there was any quality control to standardize the data collection.

Data analysis:

It is necessary to state the operational concept of SMA applied to the data collected.

Also, in general, outcomes like this, assessed by participant memory, are delimited by a recall period (e.g., last week, last month, at least once in a lifetime, etc.). In this study, a specific time period was applied?

The authors must add these information to the method, including the transcription of question(s) that assessed the SMA prevalence.

Did the questionnaire was applied in printed format? If yes, the transcription to Excel was made in duplicate? Explain if there was any quality control to reduce potential mistakes in the stage of data management.

Explanatory variables: Please present the variables investigated in the analysis (name and categories).

Results

Some results are challenging to understand if additional information is not provided to the reader, like monthly income and insurance status.

Monthly income: What is the official currency in Sudan? Did it this currency present in table 1?

Insurance status: It would be helpful to present concisely how the health system in Sudan is organized.

In table 2, the authors show the common sources and reasons behind SMA. It is important to explain how they measured these sources and reasons (closed, multiple choice question or open questions)? Were there other responses less frequent?

Based on the distribution of frequency of SMA shown in table 1, I would wait for a different pattern of association to monthly income, insurance, and level of education, at least to crude odds ratio. Please revise if the “baseline” category for these three variables is correct. Depending on this revision, the part of the discussion that comments on these results must also be revised.

Discussion

Line 199: The authors stated, "It is encouraging that the primary source of antibiotics for more than 90% of the participants was pharmacies”. It is unclear to the reader why this result is encouraging without knowing the sources available in Sudan to acquire medicines.

Also, it is essential to discuss the role of pharmacists in Sudean pharmacies since almost half of the respondents stated that pharmacists recommended antibiotics (table 6).

Minor:

Title: predicators or predictors ?

Line 38: Line 38 - Please type "World Health Organization"

Odds ratio, not odd ratio.

Author Response

Dear reviewer:

Trust this finds you well,

First of all, we thank reviewers for these comments and suggestions that lead to enhance the quality of our manuscript. We hope that our responses satisfy and address your comments. 

Reviewer 2 Report

Dear Authors,

The research article entitled "Prevalence and predicators of antibiotic self-medication in Sudan: A descriptive cross-sectional study" is interesting to read. Since there are many research articles on antibiotic self-medication which are studied across the world, authors should emphasize the novelty of the study plan. Although it is known that the study being presented was done in Sudan, were there any earlier ASM investigations undertaken there? In that case, kindly compare the results of the recent study with those from earlier ones.

Author Response

Dear reviewer:

I hope this finds you well,

we thank reviewers for these comments and suggestions that lead to enhance the quality of our manuscript. we have tried our best to explore the novelty of the work, and we have added several references to improve the comparisons. 

Reviewer 3 Report

Prevalence and predicators of antibiotic self-medication in Sudan: A descriptive cross-sectional study

It's an interesting study overall, though: It's badly written and should be reviewed by English-speaking editors. There are several methodological points to review that are included in the sectional reviews.

Introduction:

·        “Even though SM might have some positive outcomes when used properly such as reducing the cost of the treatment” – reference the argument

·        Line 49: Over Use and Miss use. They are misspelled

·        Improper use of capital letters

·        Additionally, some studies estimated that half of the antibiotics were purchased without prescription globally [10]. Recent reports indicated that non-prescription use of antibiotics reach 82% in some middle-east countries [10]. Same reference, unite the idea

·        They talk about secondary outcomes: But it is a cross-sectional study in which there is no index date, follow-up and evaluation of an outcome over time, the terminology is incorrect.

Methods:

·        The study adopted cross-sectional descriptive study design, conducted in all Sudan states 239 (all 18 states) through the period between June and December 2021. Between what days of January and December, be more specific.

·        Why was a frequency of occurrence used in the sample calculation, p= 50%?

·        How were the experts who validated the survey selected? Was any statistical test carried out for internal validation of the survey?

·        In the chi-square analysis, sociodemographic variables were evaluated against which other variables? Were these analyzes pre-specified?

·        For the logistic regression, for what reason were all the variables included in the model and criteria for including variables in the model were not considered? How did you deal with the possibility of correlation between variables? And goodness of fit?

·        How did you deal with recall bias for the participants? Of specifically remembering the self-medicated medication? Regarding the use of antibiotics previously, specifically how many days or months before, or at any time in life?

·        The questionnaire used must be known.

Results:

·        How low monthly income was defined? How the conversion to dollars was made, at what time and what exchange rate was taken into account.

·        In Sudan, how the insurance or health system works, it is worth including a short section on the topic in the introduction.

·        I did not see the distribution by region of the stratified sample, include in results.

·        Table 1: Gender, not including Male, are mutually exclusive dichotomous variables. Do not include Non-insured.

·        Line 97: X2 test results: They suggested not including them due to their unadjusted nature.

·        In Table 2: It is not clear which columns are for source of antibiotics, and which are for reason behind SMA.

·        Tonsillitis was the most common aliment? I think it's misspelled

·        I see table 3 as too loaded. I think it is important to reevaluate these crossings and leave the important ones in a simplified table.

·        Review the acronyms: COR, AOR and CI, They are not previously explained in the text.

·        Table 5: Regarding knowledge, practices and adherence, they were not previously described in methodology, Clarify. When they ask for instructions, what instructions are they talking about? In theory, if it is self-medication, nobody gives instructions.

·        Table 6: It is very striking that everyone knew the name of the antibiotic.

·        It is not reported how long ago the self-medication was.

Discussion

·        The overall discussion should be reviewed from an English writing point of view. In addition, the explanations given are superficial and should try to give new explanations to the findings and possible contribution of the study.

·        Line 196: Eretria, misspelled.

·        There is only one sentence of limitations, there are quite a few to include.

·        There is no section on specific strengths.

Author Response

Dear reviewer: 

First of all, we would like to thank you for these comments and suggestions that lead to enhance the quality of our manuscript. we have tried our best to provide comprehensive replies to your comments. 

Round 2

Reviewer 1 Report

I thank the authors for their responses, but some crucial questions remain unresolved.

Sampling technique:  I do not understand how a simple random sampling technique can apply to markets, parks, and bus stations because we need a total population to be sampled in these settings ('record of all sampling units'). It seems that a non-probabilistic sampling was adopted (e.g., convenience sampling). Please explain.

Data collection instrument

The question presented in the supplementary material ('In the last 12 months have you ever taken medicines to treat yourself without a prescription from a health professional?") gives a prevalence of the use of any medicines, not only antibiotics, right? What is the prevalence of antibiotics or any medicines? This is crucial to estimate the actual SMA prevalence, the primary outcome of this manuscript.

Table 2: The absence of option 'other' (other sources, other reasons) in the questionnaire is noteworthy. It may be a limitation of the instrument that can bias the results.

Table 4: If the distribution of SMA prevalence according to insurance status is correct in table 1, the data in table 4 remains exchanged.

Pharmacists prescription of antibiotics: In many countries, pharmacists are not authorized to prescribe antibiotics. It is unclear if pharmacists can prescribe antibiotics in Sudan and under what conditions. Please give more information to the reader to understand this issue better.

Author Response

Dear reviwer,

We again thank you for your time and valuable comments. 

Reviewer 3 Report

Most of the doubts were resolved

I still believe that the tables contain too much information and tend to be somewhat repeated, however, the authors insist on the importance of presenting all the data.

Has the style edition of the writing in English been validated, in a certified way? 

Author Response

Dear reviewer,

we thank you for your time, understanding, and your comments.
